# Effects of Electrical Stimulation on Articular Cartilage Regeneration with a Focus on Piezoelectric Biomaterials for Articular Cartilage Tissue Repair and Engineering

**DOI:** 10.3390/ijms24031836

**Published:** 2023-01-17

**Authors:** Zhengjie Zhou, Jingtong Zheng, Xiaoting Meng, Fang Wang

**Affiliations:** 1The Key Laboratory of Pathobiology Ministry of Education, College of Basic Medical Sciences, Jilin University, Changchun 130021, China; 2Department of Histology & Embryology, College of Basic Medical Sciences, Jilin University, Changchun 130021, China

**Keywords:** electrical stimulation, articular cartilage regeneration, piezoelectric biomaterials, articular cartilage tissue engineering, mesenchymal stem cells

## Abstract

There is increasing evidence that chondrocytes within articular cartilage are affected by endogenous force-related electrical potentials. Furthermore, electrical stimulation (ES) promotes the proliferation of chondrocytes and the synthesis of extracellular matrix (ECM) molecules, which accelerate the healing of cartilage defects. These findings suggest the potential application of ES in cartilage repair. In this review, we summarize the pathogenesis of articular cartilage injuries and the current clinical strategies for the treatment of articular cartilage injuries. We then focus on the application of ES in the repair of articular cartilage in vivo. The ES-induced chondrogenic differentiation of mesenchymal stem cells (MSCs) and its potential regulatory mechanism are discussed in detail. In addition, we discuss the potential of applying piezoelectric materials in the process of constructing engineering articular cartilage, highlighting the important advances in the unique field of tissue engineering.

## 1. Introduction

Articular cartilage injury is an extremely long-lasting clinical orthopedic disease that may affect function in individuals of all ages [1]. It currently affects over 50 million American adults, with that figure expected to rise to approximately 67 million by 2030 [2].

Articular cartilage is a kind of cartilage tissue with a complex morphology and strong mechanical durability [3]. Articular cartilage provides significant load-bearing and low friction properties, allowing for the smooth movement of arthrodial joints [4]. Human cartilage is divided into elastic cartilage, fibrocartilage, and hyaline cartilage according to the composition of the matrix [5]. The primary function of articular cartilage is to provide a smooth, lubricated surface for the joints and to provide a low friction coefficient to reduce friction during joint movement [6]. Because articular cartilage has no blood vessels, lymphatic vessels, or nerves, it is vulnerable to harsh biomechanical conditions [7]. Most importantly, its natural healing and repair abilities are limited and, if damaged, can lead to irreparable defects. Due to the unique and complex structure of articular cartilage, it is difficult to treat and repair the damage.

Microfracture (MF) [8], autologous chondrocyte transplantation [9], and allograft or autologous cartilage transplantation [10,11] are the currently available cartilage repair treatment options. Although these treatments have been widely used in clinical settings, there are still many shortcomings. For example, MF often leads to the formation of fibrocartilage [12]. On the other hand, donor shortage, poor integration, surgical infection, and other factors make allograft or autologous cartilage transplantation impractical [13,14]. Recently, tissue engineering has emerged as a promising strategy for repairing damaged tissue, such as skin, heart, and bone [15].

Tissue engineering is an interdisciplinary field that uses biomaterial scaffolds, seeding cells, and regulators to create biological replacements to restore, maintain, or improve tissue function. Scaffolds and regulatory factors play an important role in inducing cell differentiation and generating the desired tissue types [16]. During the construction of tissue-engineered articular cartilage, scaffolds and regulatory factors should mimic their conditions in vivo, where factors regulating cell behaviors, such as proliferation and differentiation, include not only chemical factors but also physical factors [17].

For instance, the physiological electric field (EF) plays an important role in cell activity during development [18]. Thus, electrical stimulation (ES) may be a potential regulatory factor for increasing cell proliferation, differentiation, and, eventually, ECM synthesis in vitro [19].

Bioelectrical signals spread throughout the body, and cartilage responds more strongly to ES than other tissues [20]. ES has been shown to promote cell proliferation and the production of chemicals associated with the articular cartilage ECM, such as collagen type II, aggrecan, and glycosaminoglycans (GAGs) [21]. Subsequently, direct current (DC) EFs have been used to stimulate cartilage repair [22], and biphasic currents have been shown to repair hyaline cartilage in male rats [23].

However, current ES devices that use DC contacts or capacitive coupling have problems, such as a high risk of infection, possibly painful implantation, and stress caused by surgery. These problems prevent electrical stimulators from being used in the clinic [24].

It has been shown that piezoelectric materials can change their physical properties to generate electrical signals without an external power [25]; piezoelectric materials can generate electrical signals in response to normal body movements, such as squeezing, twisting, or stimuli from outside the body (vibration, ultrasound (US), etc.) [25,26,27]. As a result, a piezoelectric scaffold can be implanted in the joint and generate ES when the joint moves. This could create an electrical stimulator that helps cartilage regrow without the need for batteries [28].

In this review, we first summarize the pathogenesis of articular cartilage injuries and the current clinical treatment strategies. We then focused on the application of ES in the repair of articular cartilage. The ES-induced chondrogenic differentiation of MSCs and its potential regulating mechanisms are discussed in detail. Furthermore, we discuss the potential of the application of piezoelectric materials in the process of constructing engineering articular cartilage, highlighting the important advances in this unique field of tissue engineering.

## 2. A Basic Outline of the Pathogenesis and Treatment of Articular Cartilage Injury

Articular cartilage is a dense connective tissue composed mainly of cells and ECM. Cells are composed of both immature and mature cell types. Immature cells in cartilage are chondroblasts, which have the ability to proliferate and secrete ECMs and further differentiate into mature cells such as chondrocytes. Cartilage’s ECM is composed of collagen, proteoglycan, noncollagen, and tissue fluid. Collagen is a strong and flexible structure that resists tension. Articular cartilage has neither nerves nor blood vessels, and its nutrition is mainly supplied by the synovial fluid and the artery branches around the synovial layer of the articular capsule. In between the collagen fibers, chondrocytes are scattered. These chondrocytes maintain the normal metabolism of articular cartilage. Articular cartilage is not only smooth but also elastic, and it can maximize absorption and buffer stress. After the injury of articular cartilage, the absorption effect of the force is reduced, and the injury and degeneration of the articular cartilage will be progressively aggravated.

Damage to the articular cartilage is a possible cause of serious musculoskeletal problems. Surface damage to cartilage disrupts cartilage mechanics and accelerates the wear process. By inducing a pro-inflammatory and catabolic state, this process makes cells more susceptible to death and causes joint damage [29]. Meantime, articular cartilage has poor inherent healing capabilities, and the injury of this tissue often leads to osteoarthritis [30]. Here, we provide a summary of the pathological mechanism of articular cartilage injury that has been elucidated by current studies.

### 2.1. Pathogenesis of Articular Cartilage Injuries

One of the causes of articular cartilage breakdown is aging. Some studies have suggested that the following factors may contribute to articular cartilage damage with age: (1) Age-related inflammation—articular cartilage injury is associated with low-grade systemic and local inflammation [31]. Aging is also associated with chronic low-grade inflammation, sometimes referred to as “inflammaging”, which could promote articular cartilage injuries, although studies to date have not identified the precise mechanisms [32]. (2) Cellular senescence—cellular senescence is one of the hallmarks of aging, and chondrocytes have many features that are characteristic of senescent cells during articular cartilage injuries [33]. (3) Mitochondrial dysfunction and oxidative stress—mitochondrial dysfunction is a hallmark of aging and is of particular concern in the context of articular cartilage injury. The free radical theory proposes that cellular damage due to the fact of high levels of reactive oxygen species (ROS) contributes to the development of the aging phenotypes and the progression of age-related diseases to a large extent [34]. In addition to cellular damage, elevated ROS levels caused by age-related oxidative stress also contribute to disease by interfering with homeostatic physiological cell signaling [35]. (4) Altered cell signaling due to ECM changes associated with aging—consistent with the concept that age-related oxidative stress alters cell signaling, studies have demonstrated that an age-related disruption of insulin-like growth factor 1 (IGF-1) signaling in human chondrocytes leads to decreased ECM gene expression and protein synthesis [36]. In human chondrocytes from the elderly, this effect is associated with increased sensitivity to oxidative stress, which leads to the inhibition of the IGF-1-mediated activation of RACα serine/threonine-protein kinase (AKT) and the increased activation of the catabolic mitogen-activated protein kinase (MAPK) signaling pathway [37,38]. Articular cartilage injuries may also be associated with local inflammation [32]. However, the clear repair mechanism needs to be further elucidated.

On the other hand, articular cartilage is a highly specialized connective tissue that is easily damaged by a variety of cytokines and chemokines [39]. Since cartilage is an avascular tissue, cartilage cells live in a hypoxic environment. Oxygen and nutrients are supplied by the vasculature of the joint capsule, synovium, and chondral bone. It has been demonstrated that a hypoxic environment has a protective effect on cartilage, because the synthesis and release of matrix metalloproteinase 1 (MMP-1) and MMP-13 and the production of type II collagen (COL2α1) cleavage fragments are lower in hypoxia than in nontoxic environments [40]. Another study indicated that articular cartilage damage was associated with an increased proteoglycan content of MMP-3 [41].

Furthermore, innate immune cells (NK cells, macrophages, and mast cells) play the most important pathogenic roles in the early inflammatory response [42]. Macrophage-derived inflammatory factors (such as tumor necrosis factor (TNF-α) and interleukin (IL-1β)) promote cartilage resorption by increasing the production of matrix-degrading enzymes [43]. These studies on the pathogenesis provide a basis for the clinical treatment of articular cartilage injuries.

### 2.2. Strategies for the Clinical Treatment of Articular Cartilage Injuries

Cartilage defects can be simply classified into three types according to the depth of the lesions: partial-thickness chondral defects, full-thickness chondral defects, and osteochondral defects [44]. Clinical approaches to the treatment of articular cartilage injury range from conservative care aimed at reducing discomfort and preserving mobility to surgical replacement of the damaged tissue (Figure 1). Bone marrow stimulation, autologous cartilage grafts, and allogeneic osteochondral grafts are all surgical interventions [45].

#### 2.2.1. Microfracture

In 1959, Pridie described the treatment of bone marrow stimulation [46]. Marrow stimulation techniques rely on the recruitment of bone-marrow-derived MSCs to the injury site. The recruited MSCs are retained within bone marrow clots, where they proliferate and differentiate into chondroprogenitor cells and produce fibrocartilage repair [47]. The microfracture technique, popularized by Steadman et al., refined marrow stimulation surgery to facilitate contact with bone marrow elements while minimizing architectural disruption of the subchondral bone [8]. Bone marrow stimulation may help relieve the symptoms of joint pain in patients [48]. It has been demonstrated that the repair tissue generated after bone marrow stimulation is mainly applied to fibrocartilage rather than hyaline cartilage [49]. Compared with other methods, MF has a definite short-term efficacy, low cost, and easy operation, but the long-term therapeutic effect is uncertain. There is evidence that retrograde changes in the subchondral region may occur after MF, such as cysts, excessive bone growth, and osteophyte formation, which would limit its application [48].

#### 2.2.2. Autologous Chondrocyte Implantation (ACI)

ACI is a procedure often used to repair large areas of damaged cartilage. ACI can be accomplished by the single implantation of chondrocytes amplified in vitro [50]. Chondrocytes can also be plated onto bioactive scaffolds before being transplanted into cartilage lesions [17]. However, the source and limited lifespan of chondrocytes limit their clinical application. On the one hand, it is difficult and expensive to obtain chondrocytes from healthy cartilage and expend them in vitro [51,52]. On the other hand, chondrocytes have a limited lifespan and potential for dedifferentiation after implantation due to the downregulation of chondrogenic markers [53]. Although the above studies show the potential of autologous chondrocytes for cartilage regeneration, the following shortcomings of ACI still limit its application [54].
Invasive surgery is required for obtaining chondrocytes;A limited number of chondrocytes are available at the donor site;The morbidity of the donor site;Loss of chondrocytes due to the fact of collagen/periosteal membrane detachment;Chondrocytes expansion in vitro is prone to dedifferentiation, and the maintenance of the chondrocyte phenotype is difficult;The proliferation and differentiation potential of autologous chondrocytes in aged patients are reduced, limiting their application;Joint replacement surgery is often unavoidable due to the generation of mechanically inferior fibrocartilage.

#### 2.2.3. Osteochondral Transplantation (OCT)

OCT can be divided into two types according to the source of the grafts. The first type is known as OAT, and the second is called OCA [54]. When compared to alternative cartilage repair techniques, OAT directly implants mature hyaline cartilage into cartilage defects, resulting in faster healing [55]. Due to the limited supply of grafts, this therapy can only be used in regions with less than 2 cm^2^ of articular cartilage damage [56]. OCA solves the problem of an insufficient number of grafts, but it is still difficult to match the allografts with the natural structure and make the fresh tissue live longer and be healthier [57]. Furthermore, allogeneic osteochondral grafts are limited by the donor source, difficulties in preserving the allogeneic tissue, and the risk of immune rejection and disease transmission [44].

#### 2.2.4. Particulate Articular Cartilage Implantation (PACI)

PACI is a process in which the articular cartilage of the same person or from different people is broken down into small pieces with a size of 1–2 mm and then put into the articular cartilage defect to promote cartilage regeneration [58]. Within 2 years of the implantation, the repair effect was good, and the repaired tissue was mostly a mixture of hyaline cartilage and fibrocartilage [59]. In theory, PACI requires less donor cartilage, thereby reducing the pain and discomfort for the donor [54].

#### 2.2.5. Nonoperative Conservative Treatment

##### Viscosupplementation with Hyaluronic Acid

Hyaluronic acid (HA) is a mucopolysaccharide naturally synthesized by chondrocytes and synoviocytes [60,61]. HA exhibits mechanical and biological properties that bind to various cell receptors and proteins, while providing lubricative properties for synovial fluid [62]. HA represents a natural therapeutic of interest given that arthritic joints display an up to 50% reduction in the HA molecular weight and concentration. The Osteoarthritis Research Society’s international guidelines state that the evidence for intra-articular HA hip injection is insufficient and should not be considered standard of care [63].

##### Intra-Articular Platelet-Rich Plasma (PRP)

PRP is an autologous centrifuged injection in which the platelet concentration is higher than for normal blood [64]. Most clinical guidelines do not recommend PRP for the treatment of osteoarthritis due to the lack of high-quality evidence of its efficacy on symptoms and joint structure, but the guidelines emphasize the need for rigorous studies. Nevertheless, PRP is growing in popularity, with many indications described and ongoing studies on efficacy. There are great differences in PRP preparation methods and inherent differences in the PRP blood/plasma content among patients [65]. Another issue is the platelet activators, which are necessary for platelet degranulation and the release of growth factors [66].

#### 2.2.6. Physical Therapy

The application of physical energy, including light, electrical stimulation, magnetism, heat, force, and other natural or artificial physical factors, through nerve, body fluid, endocrine, and other physiological regulatory mechanisms, improve blood circulation at the injured site of the cartilage and relieve pain at the injured site. For example, when an electric current passes through the human body, the distribution of the ion concentration in the body changes, which affects the permeability of the cell membrane, tissue excitability, nervous system, and peripheral blood vessels, such that the pain conduction is inhibited, the pain threshold is raised, or the pain is relieved. A percutaneous electrical stimulator was used in 288 patients with osteoarthritis of the knee. Approximately 73 percent of the patients who received electrical stimulation for more than 750 h reported pain relief, and 45 percent reported reduced anti-inflammatory drug use. In addition, the therapy delays the need for surgical intervention in chronic patients [21].

Electrical and mechanical stimulation have been proposed as tissue engineering methods to improve cartilage repair. Here, we highlight the effect of electrical stimulation in cartilage repair.

## 3. ES Promotes the In Vivo Repair of Articular Cartilage

Electromechanical transduction occurs naturally in cartilage tissue, which is based on the flow and diffusion potentials, just like in bone tissue [67,68]. When a joint moves or bears weight, the fluid flows through ionized macromolecules anchored in cartilage tissue. This will eventually create an electrical potential. The degradation of the cartilage matrix leads to the loss of this microenvironment, resulting in disruption of the EF, which is therefore important for tissue homeostasis [69]. This implies that mechanical, electrical, and electromagnetic stimulation may play an important role in regulating hyaline cartilage under normal and pathological conditions [70]. Jahr, H. et al. proposed, for the first time, that biophysical stimulation can be used to engineer cartilage tissue to achieve the purpose of cartilage lesion repair [71].

A large number of investigations have shown that ES can promote chondrogenic differentiation and cartilage maturation [72]. Electropermeabilization affects the organization of the cytoplasmic membrane and induces transmembrane ion transport, resulting in increased intracellular Ca^2+^, calmodulin activation, and TGF-β mRNA expression [73]. ES also increased the expression of the cartilage markers COL2α1, aggrecan, and Sox9, as well as decreased the expression of type I collagen. These regulatory effects of ES induced the differentiation of MSCs toward hyaline chondrocytes [74].

Furthermore, ES increases the secretion of numerous growth factors in different cells and plays a role in the chondrogenesis of MSCs. As a result, it possesses a strong therapeutic potential for cartilage regeneration. We summarized the therapeutic effects of ES, including electric and electromagnetic field stimulation, on articular injury in different animal models (Table 1). This makes it possible to determine the best treatments that might help with articular cartilage regeneration.

## 4. ES Promotes Chondrogenic Differentiation In Vitro

It has been well documented that ES promotes the proliferation of chondrocytes and the synthesis of ECM molecules, thus accelerating the healing of cartilage defects [83]. Here, we included EFs and MFs or electromagnetic fields (EMFs) into the categories of ES. Since EFs and MFs are linked and unified, they can both be transformed into each other through “movement and change”. Changing EFs with moving currents and charges produces MFs and moving MFs produces EFs as well [84]. One study indicated that DC Efs increased the expression of cartilage matrix proteins and directed the differentiation of MSCs into chondrocytes [85].

MFs or EMFs can also regulate osteogenesis and chondrogenesis [86]. For example, low frequencies of EMF (15 Hz, 5 mT) resulted in the high expression of COL2α1 and GAGs contents in bone marrow mesenchymal stem cells (BMSCs). This suggests that EMF has the potential to stimulate and maintain cartilage formation in BMSCs [87]. In addition, the scientific basis of the stimulation of bone healing by the Pain E-motion Faces Database (PEMFs) lies in the relationship between the electrical activity of the applied mechanical load and bone formation [88].

Mayer-Wagner, S. et al.showed that EMF-stimulated MSCs could also be used in clinical settings after the implantation of chondrogenic regenerates [87]. EMFs may also have beneficial effects on a variety of clinical applications [89]. In human cell models, EMF also significantly increased the cartilage markers GAGs and COL2α1. In this case, optimization parameters, such as frequency, intensity, and exposure time, must be accomplished in a detailed and precise manner [90].

ES affects the cell fate of stem cells (Table 2). Consequently, the application of ES in cartilage tissue engineering has yielded excellent results. For future clinical applications, more studies are needed to understand the signaling pathways involved in ES-regulated chondrogenic differentiation.

## 5. Possible Molecular Mechanisms Involved in ES-Promoted Cartilage Injury Repair and Cell Differentiation

It has been shown that ES can not only induce the production of chemicals involved in the repair of articular cartilage injury but also regulates proliferation/differentiation of MSCs [93]. MSCs play a central role in the natural events leading to cartilage tissue formation and repair because of their biological characteristics and differentiation potential [94]. Therefore, the molecular mechanisms of ES-regulated proliferation/differentiation of MSCs have been extensively investigated. Here, we highlight the potential signaling pathways involved in ES-regulated chondrogenic differentiation of MSCs (Figure 2).

ES is a very effective stimulator to promote cartilage growth and repair. ES initiates cell differentiation by activating calcium transients to activate calmodulin and its downstream signals [95]. Studies have shown that ES reduces the level of type I collagen and increases the expression of chondrogenic markers, such as COL2α1, aggrecan, and Sox9 SOX9, thus promoting chondrogenic differentiation [96]. Ca^2+^ enters the cell primarily through voltage-operated calcium channels (VOCCs). Several studies have shown that the inhibition of VOCCs in stem cells leads to poor chondrogenesis, especially in the early stages [97]. This highlights an important role for VOCCs in cartilage development [98].

ES can not only activate ion channels but also some voltage-sensitive genes or receptors on cell membranes, thus initiating downstream signaling pathways and generating various biological reactions [84], mediating MSC condensation and chondrogenesis. For example, ES induces chondrogenesis by promoting BMP expression, and the inhibitor of BMP signaling noggin suppressed ES-driven increases in COL2α1, AGC, and Sox9 mRNA expression [74]. During the process of ES-promoted chondrogenesis and differentiation, MAPK and Wnt signals have also been shown to be key signaling pathways [99]. Furthermore, ES pretreatment promoted chondrogenic differentiation of the MSCs by activating JNK and CREB and subsequently activating the downstream STAT3 signaling [100]. In addition, intracellular calcium may play a role in the activation of the JNK signaling pathway, suggesting that there may be crosstalk between the two signaling pathways.

Another mechanism necessary for the growth of chondrocytes is Wnt/β-catenin signaling [101]. When Wnt ligands are activated, they work with frizzled receptors and co-receptors to promote β-catenin accumulation. Once transferred into the nucleus, β-catenin regulates the transcription of genes, such as c-JNK and cyclin D1, to promote chondrocyte proliferation [102].

Other receptors on cell membranes such as purinergic receptors are activated by purines or their nucleotides. In particular, P2X_4_ has been shown to control Ca^2+^ entry into the cytoplasm in chicken mesenchymal cells, which is associated with cartilage formation [103] and promotes the chondrogenesis process by increasing the Ca^2+^ concentration [98]. Previous studies have also suggested that ES may induces TGF-β signaling, trigger the activation of P2X_4_, and leads to MSC condensation, suggesting that P2X_4_ signaling mediates the differential role of TGF-β and BMP signaling in chondrogenesis.

TGF-β can alter activin receptor-like kinase (ALK-5) and then turn on Smad 2/3/4, making COL2α1, Sox9, and aggrecan more easily generated. This will eventually lead to chondrogenic differentiation [104]. Subsequent studies further demonstrated that the BMP signaling pathway is also involved in ES-induced chondrogenic differentiation through phosphorylation of Smad 1/5/8 [105].

## 6. Application of ES on Cartilage Tissue Engineering

### 6.1. Three Main Categories of Cartilage Tissue Engineering

It is widely accepted that engineered cartilage tissue can be used as a substitute for therapeutic grafts. Engineered cartilage tissues are created by three-dimensional (3D) stem cell cultures, and these stem cells can be induced to differentiate into chondrocytes and self-assemble into engineered cartilage tissue under certain conditions [106]. Research has shown that engineered cartilage tissue can be stimulated to produce functional intermediary tissues first. When these tissues are transplanted, they continue to grow into complete cartilage tissue [107]. This suggests that chondrogenesis in vitro may be similar to chondrogenesis in vivo, including the unique cellular cascade of proliferating chondrocytes, proliferating prechondrocytes, and hypertrophic chondrocytes that assemble into a top-down multilayered structure to form regionally complex cartilage tissue [108]. In general, the elements that comprise ideal engineered cartilage tissue include seed cells, bioscaffolds, and microenvironments.

The seed cells used in cartilage tissue engineering include (1) a population of cells present in the host tissue, such as chondrocytes, chondroprogenitor cells, and osteoblasts; (2) stem cells with pluripotency; (3) induced pluripotent stem cells (iPSCs) [44]. These seed cells, either alone or in combination with other cell types [109,110], can be embedded into scaffolds to develop engineered osteochondral tissue.

Scaffolds used in cartilage tissue engineering can be classified into hydrogels [111], ceramic-bonded polymers [112], and native cartilage ECM [113]. The general properties of the matrix may affect the proliferation, migration, or differentiation of seed cells and, thus, the regeneration of articular cartilage. The chemical qualities, mechanical properties, porosity, roughness, stability, biocompatibility, bioactivity, biodegradability, and manufacturing techniques should all be considered when designing an optimal scaffold [114]. For example, successful biological scaffolds should allow for excellent cell implantation and growth of seed cells and should not exhibit toxic or inflammatory reactions upon implantation. It also has a high enough porosity to provide some interconnectivity, as well as a large surface area and sufficient ECM assembly space. In addition, biological scaffolds should be repeatable to form 3D shapes. In general, 3D scaffolds/tissue structures should ensure good cellular viability by providing essential nutrient and oxygen delivery and removing metabolic waste [115]. More interestingly, after the seed cells differentiated into chondroblasts and chondrocytes in the 3D tissue, they began to secrete cartilage substrates, such as collagen and proteoglycan, which enabled the new cartilage tissue to have structural and physical properties similar to those in vivo through self-assembly of cartilage substrates. For example, engineered articular cartilage has been able to reach the range of natural tissue in terms of the physical tensile properties (0.3–10 MPa) [116].

Because of the positive effects of ES on cartilage differentiation and cartilage injury repair, which we discussed above, ES has become an important regulatory factor and induces an ideal microenvironment in the construction of tissue-engineered cartilage. Here, we mainly discuss the scaffolds suitable for the application of ES to construct cartilage tissue engineering.

### 6.2. Conductive Scaffolds

Conductive biomaterials are a member of new generation of “smart” biomaterials that can deliver electrical, electrochemical, and electromechanical stimuli directly to cells [117]. The preparation of electroactive materials and the appropriate substrates for cell adhesion and growth can stimulate cell activity through electrical transfer [118]. Substrate conductivity can be adjusted by synthetic methods, thus affecting the physical properties, cell behavior, and tissue regeneration rate of scaffolds [119]. Conductive polymers are easy to modify and can be customized for various properties of scaffolds, making them an attractive choice for conductive components in electroactive scaffolds [120]. Conductive polymers (CPs), such as hydrogels, polyaniline (PANi), poly(3,4-ethylenedioxythiophene) (PEDOT), and polypyrrole (PPy), have common properties that are required for tissue engineering and regenerative medicine applications, such as electroactivity, reversible oxidation, hydrophobicity, biocompatibility, and surface topography.

#### 6.2.1. Hydrogels

A hydrogel is a state of material formed by hydrophilic polymers through a crosslinking network of hydrophilic macromolecules, and physical or chemical crosslinking exists between the hydrophilic macromolecules [121]. Hydrogels have excellent biocompatibility, high water content, porous structure, and adjustable mechanical strength, which are suitable for simulating ECM in tissues. Hydrogels have great potential as tissue engineering scaffolds for electroactive tissues. Hydrogels can be used as a bridge to enhance the communication and electrical coupling between normal and damaged tissues, facilitating the transmission of electrical signals in electroactive tissues [122]. The implantation of hydrogels with appropriate conductivity can effectively reconstruct physiological functions related to electrical conduction, such as cardiac pulsation, neurological conduction, and skeletal muscle contraction [123]. The conductive microenvironment provided by hydrogels can enhance the electrical communication of endogenous or exogenous cells, thus promoting the regeneration of damaged tissues [124].

#### 6.2.2. Polyaniline

PANi is another best-determined CP that has a variety of structural appearances, proper environmental durability, and facility of charge transfer by the “doping/dedoping” procedure [125]. In CPs, PANi and its derivatives occupy an increasing share in the field of tissue engineering electroactive. This is due to the unique properties of the polymer, including facile synthesis, diverse structural forms, superior thermal stability, high environmental constancy, in vitro compatibility, comfortable material accumulation, and low cost [117]. PANi has also been exhibited to have the potency to clean insidious free radicals from the environment, being a well-used method where tissues tolerate high oxidative stress, especially post infarction [120].

#### 6.2.3. Poly(3,4-ethylenedioxythiophene)

PEDOT has recently been considered an option due to the fact of its higher antioxidant properties and higher electrical conductivity. In contrast to PPy, PEDOT maintains 89% of its conductivity under the same conditions. Its high surface area and unique structure result in a lower impedance, which improves its application in bioelectrode coatings [117]. In vitro toxicity and biocompatibility tests showed that PEDOT had no cytotoxic effects on cells [126] and supported the migration, adhesion, and proliferation of human dermal fibroblast [127]. PEDOT is utilized for neural recording applications. Compared with gold, it has a low impedance and high charge density. The nanofiber structure enhances its properties [128].

#### 6.2.4. Polypyrrole

PPy is the most common choice among other conductive polymers due to the fact of its high electrical conductivity, flexible procedure of preparation, diverse surface modifications, great environmental resistance, proper biocompatibility, strong ion exchange ability, and supporting cellular activities [129]. PPy should be doped with various anions, such as Cl^−^, Br^−^, or NO_3_^−^. When opting to dopant, it should be performed meticulously, which can affect the cell growth, proliferation, and behavior [130]. PPy is an attractive CP that has been widely investigated for its efficacy towards cell functions. Since the 1990s, PPy has been studied as a cell culture substrate in in vitro culture methods [131].

However, the prolonged degradation of CPs in vivo may result in inflammation [132]. On the other hand, the application of ES in vitro may lead to the accumulation of cytotoxic byproducts, increasing the electrical resistance and generating excess heat and an unstable stimulus. For example, continuous electrical stimulation of a DC EF can result in the releasing of cytotoxic byproducts, such as Ag^+^ or other metal ions. In the process of electrical stimulation, the change in the resistance caused by electrode oxidation also leads to the instability of the voltage intensity, which ultimately leads to the various experimental results. Complex electrical stimulation equipment and operating procedures also greatly increase the chance of contamination. Therefore, the design of in vitro ES application facilities still needs to be optimized. In this case, materials with piezoelectric properties have attracted wide attention.

### 6.3. Materials with Piezoelectric Properties for Cartilage Repair and Tissue Engineering

Piezoelectricity is the electrical polarity on the surface of a crystal due to the fact of mechanical stress. Piezoelectric materials are considered smart materials, because these materials can convert the mechanical pressure acting on them into an electrical signal (the direct piezoelectric effect) and the electrical signal into a mechanical signal (the inverse piezoelectric effect). The amount of charge produced by the force on a material is proportional to the amount of external force. The methods for applying mechanical stress include ultrasonic [133], stirring [134], friction, and extrusion [25,135].

Tissues, such as bone, cartilage, dentin, tendon, and keratin, can display direct piezoelectricity [136]. For example, piezoelectric materials with bio-electroactivity can generate charges on the stressed bone surface and restore the potential on the damaged bone tissue surface, thus promoting bone healing [137].

The main substance in cartilage, collagen, is responsible for piezoelectricity. Collagen has piezoelectric properties that respond to the electrical signals generated in cartilage tissue. These signals are transmitted through the ECM to voltage-gated channels on the cell membrane [138]. Electrolysis occurs when cartilage is mechanically compressed and twisted. As a result, articular cartilage is particularly sensitive to electrical stimuli [139].

When using implantable materials in vivo, there are many things to consider. First, they are biocompatible and nontoxic. Second, the structural design needs to accommodates the proportions of the body and the irregularity of the space. The piezoelectric characteristics of a material, on the other hand, are determined by its ability to simulate the microenvironment and generate a stimulating biological response [140]. Piezoelectric polymer materials do not require energy or power to induce transient surface charges.

The piezoelectric materials used in cartilage repair and tissue engineering can be divided into synthetic and natural piezoelectric materials according to their source and properties (Table 3). Polylactic acid (PLLA), polyvinylidene fluoride (PVDF), and poly-3-hydroxybutyrate-3-hydroxy valerate (PHBV) and their copolymers are the most common piezoelectric polymers with ideal piezoelectricity [141]. Biopolymers offer certain appealing qualities compared to inorganic materials, such as a light weight, low cost, and durability [142]. Here, we limited our discussion to synthetic and natural polymers with piezoelectricity.

#### 6.3.1. Natural Piezoelectric Polymer

Naturally derived polymeric scaffolds have a biomimetic surface, natural remodeling, and bioactivity, which can enhance the interaction with the in vivo cellular system and facilitate better biological interaction performance.

Cellulose

Cellulose is a naturally occurring underground polymer with piezoelectric properties [154]. Due to the fact of its water content, nanofiber structure, high tensile strength, and ability to simulate biological environments, it can be used for a wide range of biomedical applications. Studies have shown that fibrin also contributes to cell adhesion, especially chondrocytes, osteoblasts, endothelial cells, and smooth muscle cells [152]. Cellulose is mainly obtained from different natural sources, such as bacteria, tunicates, and plants. It consists of polysaccharide macromolecules with β-1,4-glycosidic bonds and has the characteristics of hydrophilicity, cytocompatibility, bioactivity, and optical transparency, making it suitable for a variety of medical applications, such as skin tissue repair, cortical implants, drug delivery, vascular grafts, and medical implants [155]. In particular, in the field of cartilage tissue, it is widely used due to the fact of its tunability in chemical, physical, and mechanical properties [156]. The application of cellulose, including membranes, scaffold bulk materials, coatings, nanofibers, films, and nanocrystals, have led to the development of new technology strategies [157]. The scaffolds with cellulose as the main component can promote cartilage regeneration. Different techniques can control the structural alignment of the scaffolds [158]. Because of this, it is an excellent candidate piezoelectric material for cartilage tissue engineering.

Chitin

The cuticle of crustaceans, insects, and mollusks contains chitin, a natural polysaccharide with a piezoelectric structure [147]. It is a linear positively charged polysaccharide consisting of randomly distributed N-acetyl glucosamine and D-glucosamine linked byβ-1,4glycosidic bonds. Because of its charge, chitosan facilitates the interactions with several negatively charged molecules and membranes. Chitosan is characterized by cytocompatibility, biodegradability, nontoxicity, and mucoadhesivity [159]. Chitosan has many uses in medicine, such as helping to heal wounds and delivering drugs in a controlled manner. It is suitable for cartilage regeneration by combining with other filling components [160].

#### 6.3.2. Synthetic Piezoelectric Polymer

Naturally derived polymers are more biocompatible with the in vivo system but have specific limitations, such as lacking tenability, uncontrollable degradation rate, and weak mechanical strength, which may trigger an immunogenic response and microbial contamination. In contrast, synthetically designed polymers are being granted FDA approval and are being explored for practical use in various fields [161]. Due to the fact of their simple structure, synthetic polymers have tunable physical and mechanical properties, such as crystallinity and water solubility, and provides more reproducible results due to the fact of their consistent architecture. The structural requirements of the piezoelectric polymers are (1) the presence of permanent molecular dipoles; (2) the ability to align or orient the molecular dipoles; (3) the ability to maintain alignment; (4) the ability of the material to undergo large strains when subjected to mechanical stress [162].

PLLA

PLLA is a polymer synthesized from lactic acid, and it is biocompatible and biodegradable. It has a high shear piezoelectric coefficient [163] and is available in the α-, β-, and γ-crystalline forms [164]. The β-crystalline form has been proven to have the highest polarization and excellent piezoelectric properties [165]. PLLA scaffolds have been shown to induce the formation of a cartilage-like tissue [166]. PLLA is hydrolyzed and degraded, and the byproduct is PLA, which is nontoxic and water soluble. It has been well documented that degradable PLLA can achieve rapid cartilage regeneration by consuming its piezoelectric property [167].

PVDF-TrFE

PVDF-TrFE is a very flexible and nontoxic piezoelectric copolymer. It has been employed in a number of biomedical applications, from tissue-engineered scaffolds to implantable self-powered devices [135]. In addition to being biocompatible, copolymers promote cell attachment and proliferation and also help regenerate bone, skin, cartilage, and tendons, as well as other types of tissue [145]. Piezoelectric fibers have the ability to stimulate differentiated cells to form mature phenotypes and promote stem-cell-induced tissue repair. The electrospun nanofibrous-based scaffold of PVDF-TrFE copolymer can effectively regenerate articular cartilage [5]. The piezoelectric fibers stimulate differentiable cells into a mature phenotype and have the ability to promote stem-cell-induced tissue repair [168]. Currently, increasing attention is being paid to the application of polymer blends in bone and cartilage tissue engineering. Furthermore, PVDF and PVDF-TrFE have been mixed with starch- or cellulose-like natural polymers to develop scaffold structures suitable for tissue repair and regeneration, especially for bone tissue engineering. Starch or cellulose is blended to produce a porous structure to support tissue growth [169]. Therefore, scaffolds composed of electrospun PVDF-TrFE copolymer nanofibers are very beneficial for articular cartilage regeneration [146].

PHBV

PHBV has the advantages of biocompatibility, biodegradability, and thermoplasticity, and it is becoming increasingly important in the field of biomedicine. It has been reported that collagen-PHBV matrices for cartilage tissue engineering are biocompatible and take longer to break down [148]. PHBV is degraded by an enzymatic degradation mechanism, followed by hydrolysis and the release of carbon dioxide. Biodegradable PHBV-HA composites have been used in cartilage tissue engineering [167].

#### 6.3.3. Composite Piezoelectric Materials

The composite material is a biological scaffold that is formed by mixing piezoelectric materials with different properties. The physical properties of the piezoelectric properties, flexibility and biodegradability, are more consistent with the characteristics of cartilage. The main limitation of the application of piezoelectric scaffolds in cartilage tissue engineering is the nonbiodegradability of most of the piezoelectric biomaterials, including piezoelectric ceramics and PVDF-TrFE. Although biodegradable PLLA and PHBV are available, their poor mechanical properties and, more importantly, reduced piezoelectric coefficients hinder their use in piezoelectric applications. The existing strategy for making PVDF and PVDF-TrFE biodegradable is to mix nonbiodegradable piezoelectric materials with highly biodegradable polymers, including cellulose and starch [170]. In addition, polymeric ultrathin film is a new type of device that has emerged in recent years for a range of bioengineering applications, such as tissue engineering and regenerative medicine. They can be considered as quasi-two-dimensional (2D) structures with a sub-micrometric thickness and greater surface area. These geometrical features, combined with their customized surface chemistry, enable nanofilms to have a wide range of application [171]. In fact, they have been used as components of chemical/biosensors, nanoelectronic devices [172], superconducting electrical circuits and electrodes [173], and encapsulation systems for hydrogen storage materials. However, these structures have also recently shown great potential in the biomedical field [174].

#### 6.3.4. Piezoelectric Scaffolds for Cartilage Tissue Engineering

As discussed above, the discovery of endogenous EF, piezoelectricity, and their biological effects in cartilage tissues has promoted the research and development of methods for cartilage tissue regeneration by ES [175,176]. Different piezoelectric materials have been used to repair different kinds of tissue, where mechanical stress-induced electrical charges can help form cartilage [177]. Among the different types of piezoelectric materials, piezoelectric polymers have been shown to have simple machining, flexibility, and physical properties, making them suitable for a variety of applications. Research on the application of piezoelectric polymers in tissue engineering focuses on bone [140,178], neural [179,180], and muscle regeneration [181,182].

In the design of articular cartilage tissue repair scaffolds, piezoelectric scaffolds should have a 3D gradient structure, an appropriate number of pores, a certain degree of biodegradability, good biocompatibility, and initial mechanical strength [183]. Furthermore, flexibility, biodegradability, and piezoelectric properties must be considered in order to optimize the correct shape of the scaffolds and generate a piezoelectric response [184]. Tandon et al. developed a PVDF-based piezoelectric composite composed of PVDF and HAp to improve the osteogenic potential of PVDF fibers [185]. Jacob et al. reported the development of piezoelectric PHBV nanofibers supplemented with barium titanate nanoparticles for electrical stimulation of cartilage in vivo [186]. A different study by Sadeghi et al. reported that composite PHB fibers mixed with the natural polymer chitosan were added to improve the wettability, biodegradability, and biological performance of piezoelectric scaffolds [187].

However, to obtain more ideal piezoelectric scaffolds, a deeper understanding of the process by which piezoelectric materials interact with biological tissues is needed to properly adjust the physical, electrical, and biological characteristics of scaffolds for cartilage tissue engineering applications. Further study of the piezoelectric properties of natural chondral tissue is also important, not only to systematize the collected evidence but also to determine the ideal target piezoelectric coefficient value of the scaffold to provide appropriate electrical/mechanical stimulation [188].

## 7. Conclusions and Future Prospects

This review summarized the pathogenesis of particular cartilage injuries and the current clinical strategies for the treatment of articular cartilage injuries. We particularly focused on the application of ES on the in vivo repair of articular cartilage and cartilage tissue engineering. ES is an effective method for cartilage defects, and it can induce the chondrogenic differentiation of MSCs to construct engineered cartilage tissue in vitro. Using a combination of cells, scaffolds, and electrical stimulation to regenerate cartilage tissue can produce positive therapeutic results. The favorable therapeutic benefits of ES at the tissue level have motivated scientists to investigate the underlying biological mechanisms. We further highlight the different signaling pathways that are involved in ES-stimulated MSC proliferation and differentiation.

However, because of the differences in the experimental procedures, conditions, and biomaterials, determining the proper ES parameters leading to chondrogenic differentiation of MSCs is a problem that must be solved in advance. We thus focused on scaffolds suitable for the application of electrical stimulation, such as electrically conductive scaffolds and piezoelectric materials.

As a special type of ES, piezoelectric materials can conveniently generate and transmit bioelectrical signals similar to those of natural tissues so as to achieve the corresponding physiological functions. Piezoelectric scaffolds can be used as a sensitive electromechanical transduction system. Therefore, they are suitable for areas where the mechanical load is dominant, such as bone or cartilage tissue.

We strongly believe that the potential of implantable piezoelectric materials for cartilage repair and regeneration will open new doors for cartilage tissue engineering applications. These findings may contribute to the development of electrotherapeutic strategies for cartilage repair using MSCs.

Concerning tissue engineering therapies based on piezoelectric materials, the synergistic interactions between ES and piezoelectric physical structures should be investigated in detail. For example, biocompatible materials with different physical shapes, such as fibers, films, and the bulk phase, may have different piezoelectric properties. To this end, it is important to establish consistency between the shape of the piezoelectric materials and their biocompatibility. Lay, Ratanak, et al. provide a more professional and detailed summary and discussion in this aspect [189]. In this review, we mainly discussed the application of piezoelectric materials in cartilage injury repair and cartilage tissue engineering construction from a biological perspective.

Due to the many advantages of piezoelectric biomaterials and their interactions with living organisms, it is possible that they could be used to create human cartilage implants in the near future. In addition, further studies on piezoelectric biomaterials in large animal models will be conducted for evaluation in human patients in clinical trials. We believe that the ability of implantable piezoelectric materials to heal and regenerate cartilage will open up new possibilities for tissue engineering.

## Figures and Tables

**Figure 1 ijms-24-01836-f001:**
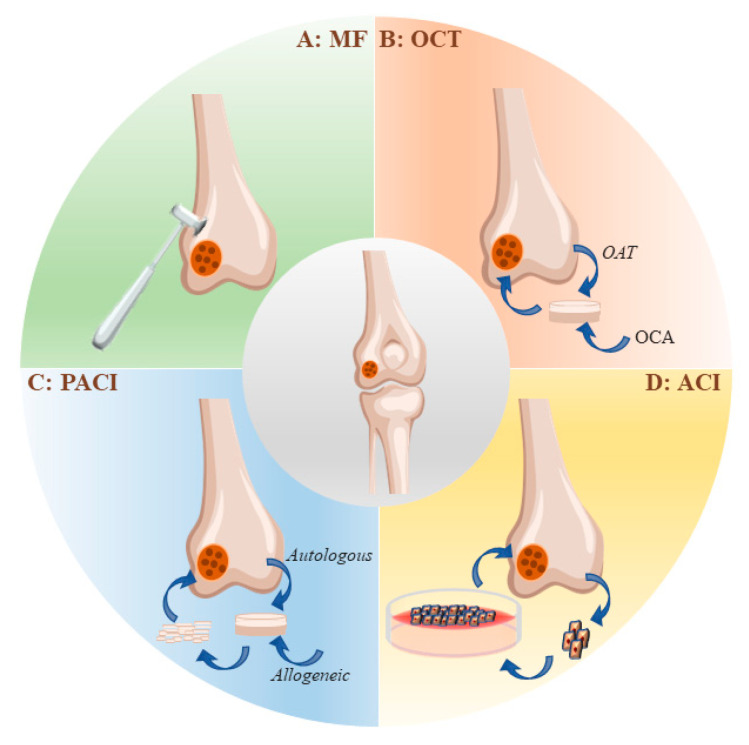
Strategies for the clinical management of articular cartilage injuries. (**A**) MF is a typical representative of bone marrow stimulation procedures. (**B**) OCT can be divided into two types according to the source of the graft. One type is known as osteochondral autologous transplantation (OAT) and the grafts are derived from the surface of the autologous non-weight-bearing joint. Another type is called osteochondral allograft transplantation (OCA) and the transplants come from the surface of the allograft joint. (**C**) Particulate articular cartilage implantation (PACI) means that articular cartilage is crushed into 1–2 mm sized particles and implanted into articular cartilage defects. PACI is divided into autologous PACI and allogeneic PACI, according to the source of grafts. (**D**) ACI refers to the collection of chondrocytes from the non-weight-bearing area of the autologous joint surface, their culture and amplification in vitro, and, finally, implantation into the cartilage defect.

**Figure 2 ijms-24-01836-f002:**
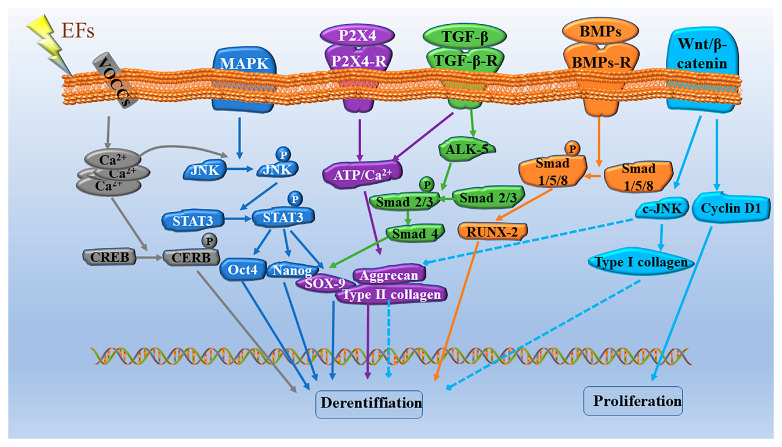
Signaling pathways involved in ES-induced proliferation and chondrogenic differentiation of MSCs. The EFs enhanced the Ca^2+^ entering the cell via VOCC, which phosphorylates CREB and, ultimately, promotes the chondrogenic differentiation of MSCs. The EFs enhanced the chondrogenic potential of MSCs via the JNK/CREB-STAT3 signaling pathway. Under ES, P2X_4_ can trigger Ca^2+^ influx into MSCs in response to interactions with ATP molecules, which contribute to the chondrogenic process. ES promotes TGF-β action on TGF-β receptors located on MSCs, thereby promoting chondrogenic differentiation. Under the action of ES, BMPs can bind to BMP receptors and promote chondrogenic differentiation. ES promotes proliferation and inhibits the chondrogenic differentiation of MSCs through the Wnt/β-catenin signaling pathway.

**Table 1 ijms-24-01836-t001:** ES promotes the in vivo repair of articular cartilage.

Type of Stimulation	Animal Model	ES Parameters	Electrode/Distance of the EFs/EMFs	Results	Reference
EFs	Femoral condyles of New Zealand white rabbits.	70 mV; 1–9 weeks	Bimetallic silver platinum electrochemical device; 3 mm silver electrodes.	Increase in cellular response, proliferation, and matrix production.	[75]
	Epiphyseal plate from rabbit femur.	1.5 V; 20 μA; 6 weeks	Two 8 cm long, twisted stainless-steel wire electrodes.	Experimental animals showed longer and broader femur on the operating side after surgery.	[76]
	Proximal tibia of the rabbit.	5 V; 60 kHz	Fitted with 1.8 × 1.8 cm stainless-steel capacitor plates over the right proximal tibial growth plate.	Small amount of electric current stimulates the epiphyseal plate to accelerate bone growth; distribution of current results in regular growth of epiphyseal.	[77]
EMFs	Knee of Hartley guinea pigs.	1.5 Hz; 1 h/day for 6 months	The applied magnetic field consisted of a pulse-burst of a 4.5 ms duration repeated at 15 burst and with a peak magnetic field of 16 G.	EMF retarded the development of osteoarthritic lesions; an increased amount of cartilage ECM, chondrocyte hypertrophy, and calcification.	[78]
	Joint knee from Hartley guinea pigs.	75 Hz; 1.5 mT; 6 h a day for 3 months	Magnetic field of 20 G, pulsed period of 67.1 ms.	Cartilage thickness (CT) was significantly higher (*p* < 0.001) in the medial tibia plateaus; significant reduction of chondroplasty progression.	[79,80]
	Distal femoral growth plates of male Wistar rats.	110 Hz; 2 mT; 2 h/day for 90 days	A 0.2 mm copper wire, 0.5 cm internal diameter of the coil probe.	Rats treated with ES experienced more rapid weight gain; chondrocytes rapidly proliferated, matured, and transformed into hypertrophic cells in the growth plate calcium; growth hormone levels were higher.	[81,82]

EMFs: electromagnetic fields; EFs: electric fields.

**Table 2 ijms-24-01836-t002:** ES induced stem cell differentiation towards chondrocytes in vitro.

Stem Cell Type	ES Modality	Parameters	Differentiation Medium	Results	Reference(s)
ADSCs	EFs	1 KHz, 20 mV/cm, 20 min/day, 7 days	Chondrogenic differentiation media DMEM-high glucose; penicillin and streptomycin: 1%; dexamethasone: 10^−7^ M; ascorbat-2-phosphate: 50 μg/mL; bovine serum albumin: 0.5 mg/mL; linoleic acid: 5 μg/mL; insulin: 10 mg/mL; transferrin: 5.5 mg/L; selenium (insulin-transferrin-selenium): 5 μg/L.	Increase in the expression of COL2α1 and Sox9 genes; decrease in the expression of type I and type X collagen genes.	[91,92]
BMSCs	EMFs	15 Hz, 5 mT, 45 min every 8 h	Chondrogenic differentiation media DMEM-high glucose; insulin: 10 mg/mL; transferrin: 5.5 mg/L; selenium: 5 mg/L; bovine serum albumin: 0.5 mg/mL; linoleic acid: 4.7 mg/mL; dexamethasone: 0.1 mM; L-ascorbic acid-2-phosphate: 0.2 mM; L-proline: 0.35 mM; penicillin/streptomycin: 30 U/mL; FGF-2: 5 ng/mL.	Increased in COL2α1 expression, decrease in collagen type X expression.	[87]
BMSCs	PEMFs	15 Hz, 2 mT, 10 min, daily	Chondrogenic differentiation media DMEM-high glucose; proline: 4 mM; ascorbic acid: 50 µg/mL; sodium pyruvate: 1 mM; dexamethasone: 10^−7^ M; transforming growth factor-β3 (TGFβ3): 10 ng/mL.	Moderate enhancement in the gene expression of Sox9, aggrecan, and COL2α1 and the deposition of GAGs.	[88]

ADSCs: adipose-derived mesenchymal stem cells; BMSCs: bone marrow mesenchymal stem cells.

**Table 3 ijms-24-01836-t003:** The piezoelectric materials used for cartilage repair.

Category	Material	Piezoelectric Coefficient	Material Biological Advantages	Restoration Result	Reference(s)
Synthetic piezoelectric materials	PVDF	d_31_ = 20 pC/N	Highly elastic; nontoxic; biocompatible.	Promotes cell adhesion and proliferation of chondrogenic cells.	[143,144]
	P(VDF-TrFE)	d_33_ = 30 pC/N	Cytocompatible.	Piezoelectric fibers can stimulate the differentiable cells into a mature phenotype and promote tissue repair.	[145,146]
	PHBV	d_33_ = 1.3 pC/N	Biocompatible; more extended biodegradation rate.	Formed hyaline such as cartilage and neocartilage was integrated into the adjacent cartilage.	[147,148]
	PLLA	d_14_ = −10 pC/N	Biodegradable; biocompatible; strong mechanical properties; nontoxic; water soluble.	Rapid bone and cartilage regeneration by consuming the piezoelectric property.	[149,150]
Natural piezoelectric materials	Cellulose	d_14_ = 0.2 pC/N	Excellent biocompatibility.	Offers biological signaling, cell adhesion, and remodeling.	[151,152]
	Chitin	d_14_ = 0.2–1.5 pC/N	Natural polysaccharide; hydrophilic material; biocompatible.	Carriers for controlled drug delivery; promote cell adhesion, proliferation, and differentiation, providing support for cartilage regeneration; favor integration to the subchondral region.	[153]

P(VDF-TrFE): poly(vinylidene fluoride–trifluoroethylene).

## Data Availability

Not applicable.

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
