# Peer review of "Effects of Electrical Stimulation on Articular Cartilage Regeneration with a Focus on Piezoelectric Biomaterials for Articular Cartilage Tissue Repair and Engineering"

_ijms, 2023, doi:10.3390/ijms24031836_

Round 1

Reviewer 1 Report

Comprehensive evaluation of the paper:

In this study,  summarizes the pathogenesis of particular cartilage injuries and current strategies for the clinical treatment of articular cartilage injuries, focus on the application of ES on in vivo repair of articular cartilage and cartilage tissue engineering.

The work in this paper is a reference for the treatment of clinical articular cartilage injuries, The manuscript can be considered for publication in the journal subject to a further revision.

Modification opinions on the paper:

The existing problems are as follows:

1)       The authors should have detailed the properties of the articular cartilage itself, such as its physical properties.

2)       Current treatments for articular cartilage injuries should be listed separately.

3)       Lack of current research advances in the treatment of articular cartilage injuries.

4)       Materials with piezoelectric properties should be more detailed.

Author Response

Reviewer 1

Comments and Suggestions for Authors

Comprehensive evaluation of the paper:

In this study, summarizes the pathogenesis of particular cartilage injuries and current strategies for the clinical treatment of articular cartilage injuries, focus on the application of ES on in vivo repair of articular cartilage and cartilage tissue engineering.

The work in this paper is a reference for the treatment of clinical articular cartilage injuries, The manuscript can be considered for publication in the journal subject to a further revision.

Modification opinions on the paper:

The existing problems are as follows:

  • The authors should have detailed the properties of the articular cartilage itself, such as its physical properties.

We have added more details into the structural and physical properties of the articular cartilage (line 83-94, in the text of ‘clear final version’).

  • Current treatments for articular cartilage injuries should be listed separately.

We appreciate for reviewer’s suggestions, and this is indeed a very meaningful and constructive proposal.

We have separated section2 to 2 parts with subsection 2.1 (Pathogenesis of Articular Cartilage Injuries) and 2.2 (Strategies for the clinical treatment of articular cartilage injuries). In section 2.2, we added more current treatments for articular cartilage injuries and listed them separately (line 214-247, in the text of ‘clear final version’).

  • Lack of current research advances in the treatment of articular cartilage injuries.

In section 2.2, we have added current research advances in the different treatment of articular cartilage injuries.

(4)       Materials with piezoelectric properties should be more detailed.

We agree with the referee that further discussions on the piezoelectric properties of materials should be included. We have added more details into the section 6.3. The properties, advantages and disadvantages of each piezoelectric material are further introduced and discussed.

Reviewer 2 Report

Dear authors

I have some suggestions for you.

There are some faults for example p 6 lane 228 and p8 lane 273: correct induces by induce, p11 lane 421: tisuue by tissue, lane 424: highlght by highlight, p12 lane 444: shpes by shapes. Put all in vitro in italics.

For a better understanding it would be necessary to remake tables 1, 2 and 3. Indeed the results are following each other and the references are in the middle.

Best regards

Author Response

Reviewer 2

Comments and Suggestions for Authors

  • There are some faults for example p 6 lane 228 and p8 lane 273: correct induces by induce, p11 lane 421: tisuue by tissue, lane 424: highlght by highlight, p12 lane 444: shpes by shapes. Put all in vitro in italics.

We have fully checked the errors and corrected them.

(2) For a better understanding it would be necessary to remake tables 1, 2 and 3. Indeed the results are following each other and the references are in the middle.

We have remade the tables and added the content into the Table 1, 2 and 3, which makes the table more meaningful and clearer.

  Reviewer 3 Report

The manuscript reports a review of the application of electrical stimuli, with different approaches, targeting articular cartilage regeneration.

In my opinion, the work needs several improvements before considering it suitable for publication. 

I suggest addressing the following comments:

- the title may suggest an analysis of the electrical stimulation applied even with piezoelectric materials. However, I do not see any reference on the electrical signal effectively generated by the piezoelectric components, by mechanical, ultrasound or other stimuli. I strongly suggest the authors to enrich the manuscript with a deeper description of the electrical signals (in voltage, for exampl) generated by piezoelectric elements, for example also taking into consideration the following references:

o Cafarelli, Andrea, et al. "Piezoelectric nanomaterials activated by ultrasound: the pathway from discovery to future clinical adoption." ACS nano 15.7 (2021): 11066-11086.

o Ricotti, Leonardo, et al. "Ultrasound stimulation of piezoelectric nanocomposite hydrogels boosts cartilage regeneration." (2022).

- I suggest to deepen the section "Pathogenesis of Articular Cartilage Injuries". Just a few details have been reported. Also the section "Strategies for the clinical treatment of articular cartilage injuries" is poorly detailed. Since the title comprises "clinical", there are several not mentioned approaches. (e.g., viscosupplementation)

- Table 1. Please explain the meaning of EF. And, if possible, be coherent with the parameters used in the tests. A mention of the setup employed would let the reader understand better the meaning of such studies (e.g., electrode applied outside or inside, distance of the EMFs, etc...).

- Table 2. Did the study consider also the use of a differentiation medium, or the results were the object of solely electrical stimulation?

- Please enlarge Figure 2, to make it more readeable.

- In the section 6, the authors wrote "Therefore, scaffolds used for tissue engineered cartilage should have physical properties similar to those of cartilage in vivo.". Such a statement is difficult to combine with cell-laden materials able to promote the regeneration of the tissue, since a too-rigid scaffold may hamper cell viability.

- Since the main focus is the electrical stimulation, apart from the analysis of the piezoelectric materials, the authors must report the state of the art about electro-sensitive scaffolds, which can contribute to the stimulation of the cells.

- The authors stated, "the application of ES in vitro may lead to the accumulation of cytotoxic byproducts, increasing the electrical resistance and generating excess heat and an unstable stimulus." Please enrich with more details this topic.

- Table 3. Which is the piezoelectric coefficient reported in the Table? and clarify what you intend with "Biological characteristics".

- Figure 3. What do you intend with the shaking table? In the figure, an orbital shaker is reported, but I would avoid such an image since it seems pretty unusual the use of such a "mechanical" stimulation. I suggest removing this figure since it is not informative.

- I suggest revising the part on the optimization of the piezoelectric scaffold. The information reported is too qualitative, and does not provide any useful hint for a reader, apart from the need to optimize. Some quantitative details must be provided. I also suggest adding the use of composite materials, made of polymeric matrices and piezoelectric fillers, since their role is still relevant in the electrical stimulation of the cartilage tissue. The authors should also refer to different types of scaffolds, for example, 2D thin film. Take the following reference as an example:

o Vannozzi, Lorenzo, et al. "Novel ultrathin films based on a blend of PEG-b-PCL and PLLA and doped with ZnO nanoparticles." ACS applied materials & interfaces 12.19 (2020): 21398-21410.

- Please explain the meaning of "circumstances" in line 439

- The authors reported "Concerning tissue engineering therapies based on piezoelectric materials, synergistic interactions between ES and piezoelectric physical structure should be investigated in detail. For example, the biocompatible materials with different physical shpes, such as fibers, films, and the bulk phase, may have different piezoelectric properties. To this end, it is important to establish consistency between the shape of the piezoelectric materials and its biocompatibility.". There are also some researches that highlighted interesting points on the topics touched by the authors, as the following reference:

o Lay, Ratanak, Gerrit Sjoerd Deijs, and Jenny Malmström. "The intrinsic piezoelectric properties of materials–a review with a focus on biological materials." RSC advances 11.49 (2021): 30657-30673.

Please the authors revise better this part of the conclusion in view of what is existing already in the scientific literature.

Author Response

Reviewer 3

Comments and Suggestions for Authors

The manuscript reports a review of the application of electrical stimuli, with different approaches, targeting articular cartilage regeneration.

In my opinion, the work needs several improvements before considering it suitable for publication. 

We appreciate for review’s suggestion and the manuscript has been proofread by a native speaker.

(1)- the title may suggest an analysis of the electrical stimulation applied even with piezoelectric materials. However, I do not see any reference on the electrical signal effectively generated by the piezoelectric components, by mechanical, ultrasound or other stimuli. I strongly suggest the authors to enrich the manuscript with a deeper description of the electrical signals (in voltage, for example) generated by piezoelectric elements, for example also taking into consideration the following references:

o Cafarelli, Andrea, et al. "Piezoelectric nanomaterials activated by ultrasound: the pathway from discovery to future clinical adoption." ACS nano 15.7 (2021): 11066-11086.

o Ricotti, Leonardo, et al. "Ultrasound stimulation of piezoelectric nanocomposite hydrogels boosts cartilage regeneration." (2022).

ⅰ. We have enriched the manuscript with a deeper description of the electrical signals generated by piezoelectric elements (line 472-478, in the text of ‘clear final version’).

ⅱ. We also redid the Table 3 and added description of the piezoelectric coefficient into it. The piezoelectric coefficient is the conversion coefficient of mechanical energy into electrical energy or electrical energy into mechanical energy, reflecting the coupling relationship between the elastic and dielectric properties of piezoelectric materials. The charge Q or voltage V generated by piezoelectric material under external applied force is the application of positive piezoelectric effect. Charge can be derived from the formula:

Q= F (Force) ×d33 (Piezoelectric Coefficient)

Therefore, piezoelectric coefficient rather than voltage or charge is used to evaluate the properties of piezoelectric materials.

ⅲ. We are grateful for the reviewer's suggested references and added the reference (Ref 27) into text. (we could not find full text from o Ricotti, Leonardo, et al.)

 (2)- I suggest to deepen the section "Pathogenesis of Articular Cartilage Injuries". Just a few details have been reported. Also the section "Strategies for the clinical treatment of articular cartilage injuries" is poorly detailed. Since the title comprises "clinical", there are several not mentioned approaches. (e.g., viscosupplementation)

We have separated section2 to 2 parts with subsection 2.1 (Pathogenesis of Articular Cartilage Injuries) and 2.2 (Strategies for the clinical treatment of articular cartilage injuries). In section 2.2, we added more current treatments for articular cartilage injuries and listed them separately (line 214-247, in the text of ‘clear final version’).

(3)- Table 1. Please explain the meaning of EF. And, if possible, be coherent with the parameters used in the tests. A mention of the setup employed would let the reader understand better the meaning of such studies (e.g., electrode applied outside or inside, distance of the EMFs, etc...).

We appreciate for reviewer’s suggestions, and this is indeed a very meaningful and constructive proposal. We have redone the Table1 and added a description of the type of electrical stimulation applied to the body and the placement of electrodes.

(4)- Table 2. Did the study consider also the use of a differentiation medium, or the results were the object of solely electrical stimulation?

This issue is indeed of concern, and we have added the composition of differentiated medium in Table 2 to remind the investigator to consider the possible differences in the outcome of the synergistic effect between electrical stimulation and medium composition.

(5)- Please enlarge Figure 2, to make it more readeable.

We have enlarged the Figure2 and make it more readable.

(6)- In the section 6, the authors wrote "Therefore, scaffolds used for tissue engineered cartilage should have physical properties similar to those of cartilage in vivo.". Such a statement is difficult to combine with cell-laden materials able to promote the regeneration of the tissue, since a too-rigid scaffold may hamper cell viability.

We agree with the reviewer. A too-rigid scaffold may hinder cell implantation and growth. Therefore, seed cells should first be plated in scaffolds that promote cell adhesion and growth when constructing engineered cartilage. In another hand, other physical properties should be simulating the natural cartilage tissue, such as porosity, 3D environment.

We have modified and added discussion in the text (line 387-398, in the text of ‘clear final version’):

(7)- Since the main focus is the electrical stimulation, apart from the analysis of the piezoelectric materials, the authors must report the state of the art about electro-sensitive scaffolds, which can contribute to the stimulation of the cells.

We appreciate for reviewer’s suggestion. We have added the discussion regarding electro-sensitive scaffolds in the text as section 6.2 (6.2. Conductive Scaffolds) (line 404-470, in the text of ‘clear final version’).

(8)- The authors stated, "the application of ES in vitro may lead to the accumulation of cytotoxic byproducts, increasing the electrical resistance and generating excess heat and an unstable stimulus." Please enrich with more details this topic.

We have enriched more details in the topic "the application of ES in vitro may lead to the accumulation of cytotoxic byproducts, increasing the electrical resistance and generating excess heat and an unstable stimulus." (Line 461-470, in the text of ‘clear final version’)

(9)- Table 3. Which is the piezoelectric coefficient reported in the Table? and clarify what you intend with "Biological characteristics".

We have redone the Table3 and added a description of biological advantages of piezoelectric materials.

(10)- Figure 3. What do you intend with the shaking table? In the figure, an orbital shaker is reported, but I would avoid such an image since it seems pretty unusual the use of such a "mechanical" stimulation. I suggest removing this figure since it is not informative.

We removed the figure3 from the text.

(11)- I suggest revising the part on the optimization of the piezoelectric scaffold. The information reported is too qualitative, and does not provide any useful hint for a reader, apart from the need to optimize. Some quantitative details must be provided. I also suggest adding the use of composite materials, made of polymeric matrices and piezoelectric fillers, since their role is still relevant in the electrical stimulation of the cartilage tissue. The authors should also refer to different types of scaffolds, for example, 2D thin film. Take the following reference as an example:

o Vannozzi, Lorenzo, et al. "Novel ultrathin films based on a blend of PEG-b-PCL and PLLA and doped with ZnO nanoparticles." ACS applied materials & interfaces 12.19 (2020): 21398-21410.

We appreciate for reviewer’s suggestion. In section 6.3, we divided piezoelectric materials into three categories and discussed in details:

6.3.1. Natural piezoelectric polymer,

6.3.2. Synthetic piezoelectric polymer;

6.3.3. Composite materials.

We also added content about novel types of scaffolds into the text, for example, 2D thin film (line 593-598, in the text of ‘clear final version’).

 We have added some details and the discussion on ‘the optimization of the piezoelectric scaffold’ in the text at section 6.3.4 (line 602-630, in the text of ‘clear final version’).

(12)- Please explain the meaning of "circumstances" in line 439

What we want to express here is the culture environment or condition. We have changed to a more accurate word “condition”.

(13)- The authors reported "Concerning tissue engineering therapies based on piezoelectric materials, synergistic interactions between ES and piezoelectric physical structure should be investigated in detail. For example, the biocompatible materials with different physical shapes, such as fibers, films, and the bulk phase, may have different piezoelectric properties. To this end, it is important to establish consistency between the shape of the piezoelectric materials and its biocompatibility.". There are also some researches that highlighted interesting points on the topics touched by the authors, as the following reference:

o Lay, Ratanak, Gerrit Sjoerd Deijs, and Jenny Malmström. "The intrinsic piezoelectric properties of materials–a review with a focus on biological materials." RSC advances 11.49 (2021): 30657-30673.

Please the authors revise better this part of the conclusion in view of what is existing already in the scientific literature.

We appreciate for reviewer’s suggestion. In our review, we mainly discussed the application of piezoelectric materials in cartilage injury repair and cartilage tissue engineering construction from a biological perspective. The impact of physical shape of materials on piezoelectric properties was discussed in more detail and professionally in the following reference proposed by the reviewer: ‘The intrinsic piezoelectric properties of materials–a review with a focus on biological materials’. We have pointed out in the text that this reference made wonderful summary and discussion in this aspect.

Round 2

Reviewer 3 Report

Nothing to add